# Status of Concentrated Solar Power Plants Installed Worldwide: Past and Present Data

Sylvain Rodat [1,*] and Richard Thonig [2]

1   Processes, Materials and Solar Energy Laboratory, PROMES-CNRS, 7 Rue du Four Solaire,
    66120 Font Romeu, France
2   Energy Transitions and Public Policy, Research Institute for Sustainability—Helmholtz Centre Potsdam (RIFS),
    14467 Potsdam, Germany
*   Correspondence: sylvain.rodat@promes.cnrs.fr

**Abstract:** Solar energy is not only the most abundant energy on earth but it is also renewable. The use of this energy is expanding very rapidly mainly through photovoltaic technology. However, electricity storage remains a bottleneck in tackling solar resource variability. Thus, solar thermal energy becomes of particular interest when energy storage is required, as thermal energy storage is much cheaper than electricity storage. The objective of this paper is to make a short update on the CSP (Concentrated Solar Power) market as of the year 2023. It is based on the CSP-GURU database, which lists information on CSP power plants all over the world. Although this database is open, it is not easy to find UpToDate analysis. An overview of this expanding technology is presented and offers readable figures with the most important information. This includes the evolution of installed capacities worldwide along with upcoming projects (under construction) and technological trends. The evolution of storage capacities and operating temperatures is discussed. Investment costs and levelized cost of electricity are also provided to obtain reliable data for comparison with other energy technologies. Specific land requirements are highlighted, along with overall efficiency. Relevant examples are discussed in this paper. Eventually, it outlines the evolution of the CSP landscape with useful information for scientific and educational purposes.

**Keywords:** concentrated solar power; parabolic trough; Fresnel concentrator; solar tower; technology status; LCOE



## 1. State of the Art

The most abundant energy source on Earth is renewable and comes from the sun. The use of this energy can be with two technologies: photovoltaic (PV) cells and concentrated solar power (CSP). The former directly converts photons into electricity via the photoelectric effect. The total cumulative installed PV capacity in 2023 was about 1.5 TWp [1]. However, electricity storage with batteries is still expensive, which makes it difficult to deal with solar energy variability of days and weeks. CSP, on the other hand, converts sunlight into thermal energy that can be further converted to electricity by thermodynamic cycles. Thermal energy storage provided by CSP technology is a specific asset [2]. Concentrated solar power is a way to produce heat or electricity by means of solar rays' concentration onto a receiver. At the difference of PV, it uses only the direct normal irradiance. Various approaches to concentrating sunlight exist (Figure 1) and can be categorized, depending on whether the mirrors are tracking the sun along one axis (linear technologies) or two axis (point focus technologies). Depending on whether the receiver is mobile or not, the technologies can be further categorized in parabolic trough or linear Fresnel for linear technologies and in parabolic dishes or solar towers for point focus technologies. Beam-down towers are similar to solar towers but with secondary optics aiming at redirecting the solar flux to the fixed receiver on the ground. At the focus of each technology, a receiver cooled by a heat transfer fluid is installed so that solar heat can be extracted and directed to

the turbine or other downstream processes. The main heat transfer fluids used are oil, water and molten salts, depending on the operating temperature [3]. Molten salts are mainly used as sensible storage medium either in direct or indirect configuration (depending on the heat transfer fluid is also molten salt or not). Storage takes the form of two tanks and the salt flows from the cold tank to the receiver where it is heated up, and then the hot salt is stored in the hot tank during the day. After sunset, hot salt is used to generate steam for the turbine and flows back to the cold tank. Point focusing systems can reach a concentration factor beyond 500 while linear focusing systems provide a concentration ratio below 100. The consequence is that thermal efficiency is higher with two-axis tracking systems and thus higher temperatures can be reached. A didactic paper by Steinfeld and Meier [4] is available to provide the main theoretical aspects of concentrated solar technology, while the recent papers of Islam et al. [5] and Khan et al. [6] report on the current technology development.

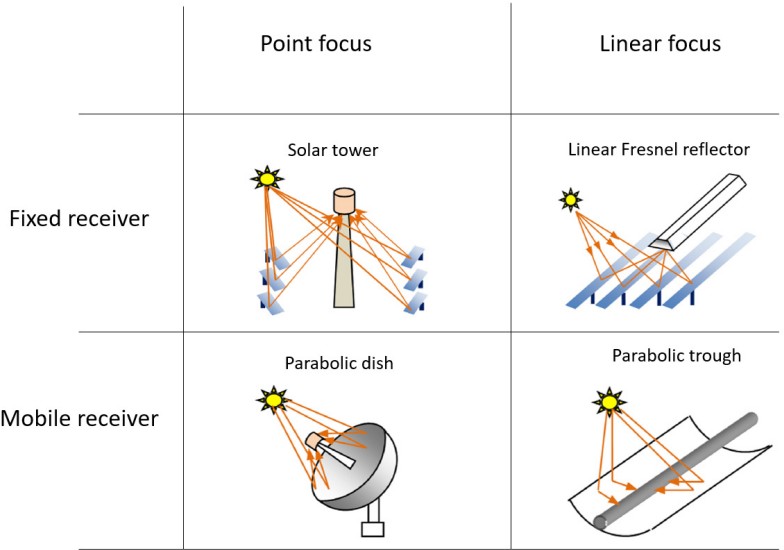

**Figure 1.** The four main CSP technologies (adapted from [7]).

The purpose of this paper is not to detail the various solar technologies (more information can be found in various references [8–10]) but to analyze concisely the past and current status of the technology. For that purpose, updated data from the CSP.guru database as of July 2023 [11] are used to plot meaningful figures with the most important information regarding installed capacities, active countries, selected technologies, storage capacities, operation temperature, investment costs, levelized cost of electricity, specific land requirements and overall efficiency. The database of CSP.guru builds on the previously existing NREL–SolarPACES database, which was based on personal correspondence. Towards this end, it collects references from several online sources, including dedicated publications on CSP, academic papers and other publicly available sources (sources are archived via the internet archive and listed in a dedicated column titled "Sources"); additionally, people directly involved with the projects through national research organizations, such as NREL in the US, ENEA in Italy or PSA in Spain, were contacted. All economic data such as LCOE have been calculated using a unified methodology [12,13]. Previous works were already published from this dataset but with a different level of analysis and older versions [12,14]. Also, a data discovery web application for commercial CSP plants is currently available but is not complete yet [15]. This work aims to give a practical overview of the current status of the technology both for scientific and educational purposes. In particular, it provides updated figures that are not available elsewhere (interactive HTML figures, improving readability, are also attached to this publication-see Supplementary Materials).

## 2. Global CSP Deployment

### 2.1. Worldwide Capacity

Figure 2 plots the total CSP capacity in operation worldwide and the main countries involved (operational capacity > 50 MW). The first solar power plant reported is the one from the US 5 MW National Solar Thermal Test Facility, in operation since 1978. Then, a long period of almost 30 years shows a very slow deployment of the CSP technology before a boom between 2008 and 2013 sustained by Spain and the USA. Within this 5-year period, about 4 GW of CSP were built. Then, other countries entered the market, such as China, Morocco, South Africa, India and Israel, to reach a total actual CSP capacity slightly higher than 6.5 GW as of July 2023.

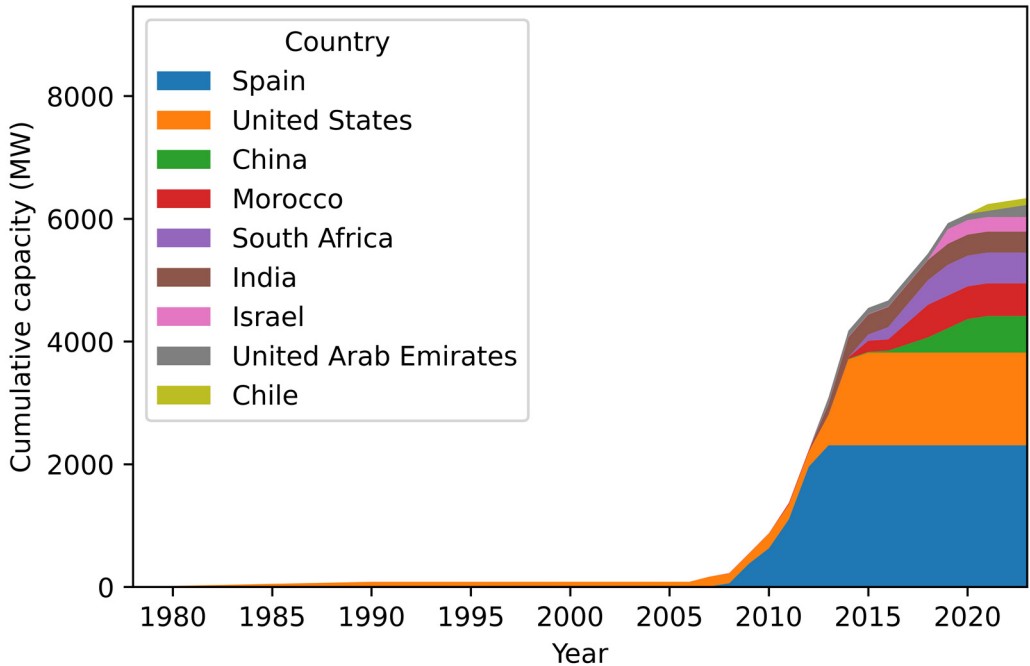

**Figure 2.** Cumulative CSP capacities around the world (plants in operation).

Figure 3 provides a list of the main countries involved in the CSP market and their respective share in the total solar thermal power. It considers both operational and under-construction capacities to provide some perspectives. Also, the CSP technologies are distinguished. All countries are sorted in the increasing order of total capacity (capacity installed or under construction). Only countries with a total capacity of over 50 MW are plotted. It shows that Spain is the leader (2300 MW operational), followed by the United States (1500 MW operational) and China (596 MW operational). These three countries account for more than 70% of the world's CSP operating capacity. However, it has to be noted that both Spain and the United States have no project under construction. New projects are mainly appearing in China and the UAE. China shows mainly projects directed toward the central tower technology (810 MW under construction). It is also the first country to scale up the beam-down technology up to 50 MW (Yumen Xinneng/Xinchen—50 MW beam down). This plant operates at temperatures between 570 °C and 290 °C with molten salts and a 9 h two-tank direct storage. Electricity is produced through a Rankine cycle at 140 bar and 43.7% efficiency. The beam down is 70 m in height. For its part, the UAE is more focused on parabolic trough technology. In fact, in December 2023, the world's largest concentrated solar power project was inaugurated in Dubai. It is composed of a 600 MW parabolic trough plant (still under construction at the date of the database), a 100 MW solar tower plant coupled with 250 MW from photovoltaic solar panels. The parabolic trough plant operates with thermal oil in the solar field and stores thermal energy in a molten salt two-tank indirect storage with 11 h of nominal production capacity. The solar tower is also

the tallest of its kind with 263 m [16]. On top of it, a molten salt receiver is used to convert solar radiation to thermal energy. The storage is a direct molten salt two-tank system with a 15 h storage capacity. These projects brought the total CSP capacity beyond 7 GW at the end of 2023.

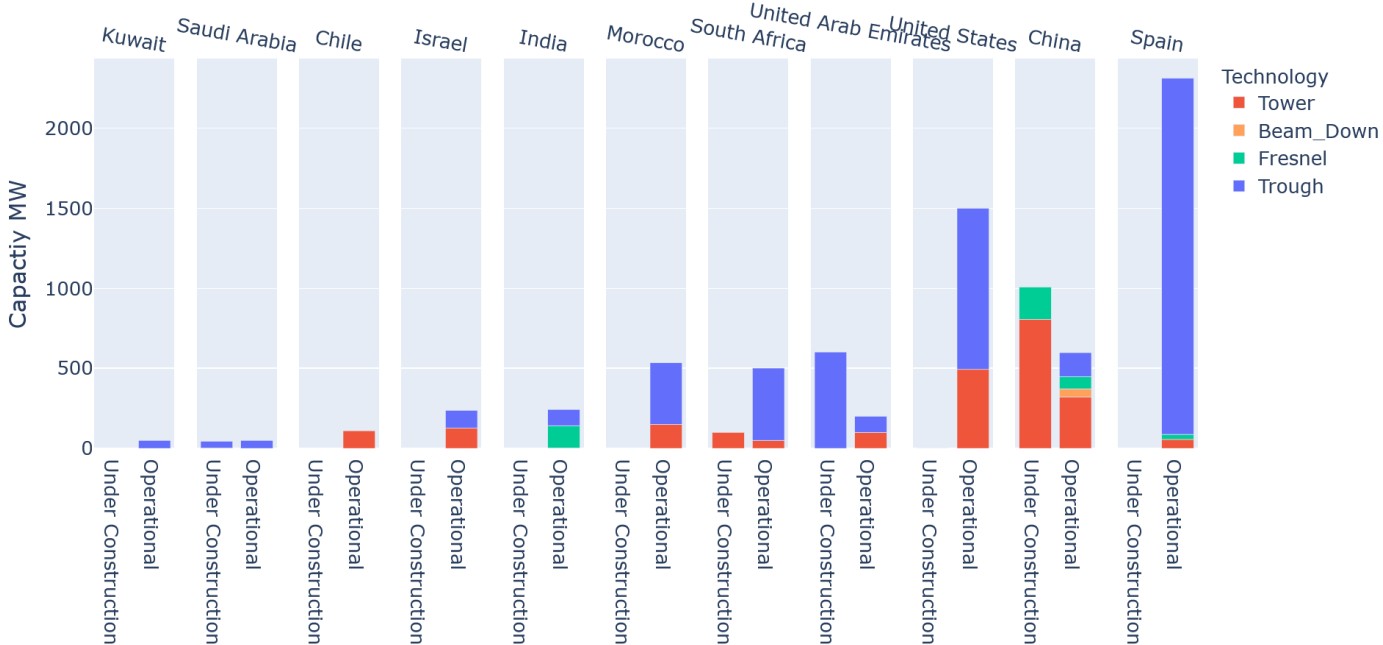

**Figure 3.** Main countries involved in CSP with their respective capacities in operation or under construction.

*2.2. Technology Repartition*

Figure 4 provides the repartition of the four CSP technologies including operational trough, tower, Fresnel and dishes since the 1990s. In 2023, parabolic trough appears as the most deployed technology (73%), followed by tower (23%) and Fresnel (4%). There is currently no dish solar power plant in operation. It has to be noted that although dishes have high efficiencies, economies of scale are limited for this technology due to the fact that the maximum dish size is limited by mechanical constraints. Historically, parabolic trough has been the first technology deployed at a large scale in Spain and the USA. Towers have gained attention after 2012 in order to increase the operating temperatures (see dedicated part) as two-axis tracking systems are optically more performant [17]. Also, it could be a more sustainable alternative to parabolic trough [18]. On the contrary, Fresnel deployment started in the meantime but remained marginal. Fresnel is expected to compete with parabolic trough as the technology is simpler (lower cost) but its optics are also slightly less efficient [19]. Solar thermal technology is also sometimes hybridized either with PV, wind or ISCC (Integrated Solar Combined Cycle). In this paper, the choice was made to dispatch hybrid plants to their respective CSP technology, so that, for example, a 100 MW Fresnel + 900 MW PV hybrid project counts as 100 MW Fresnel.

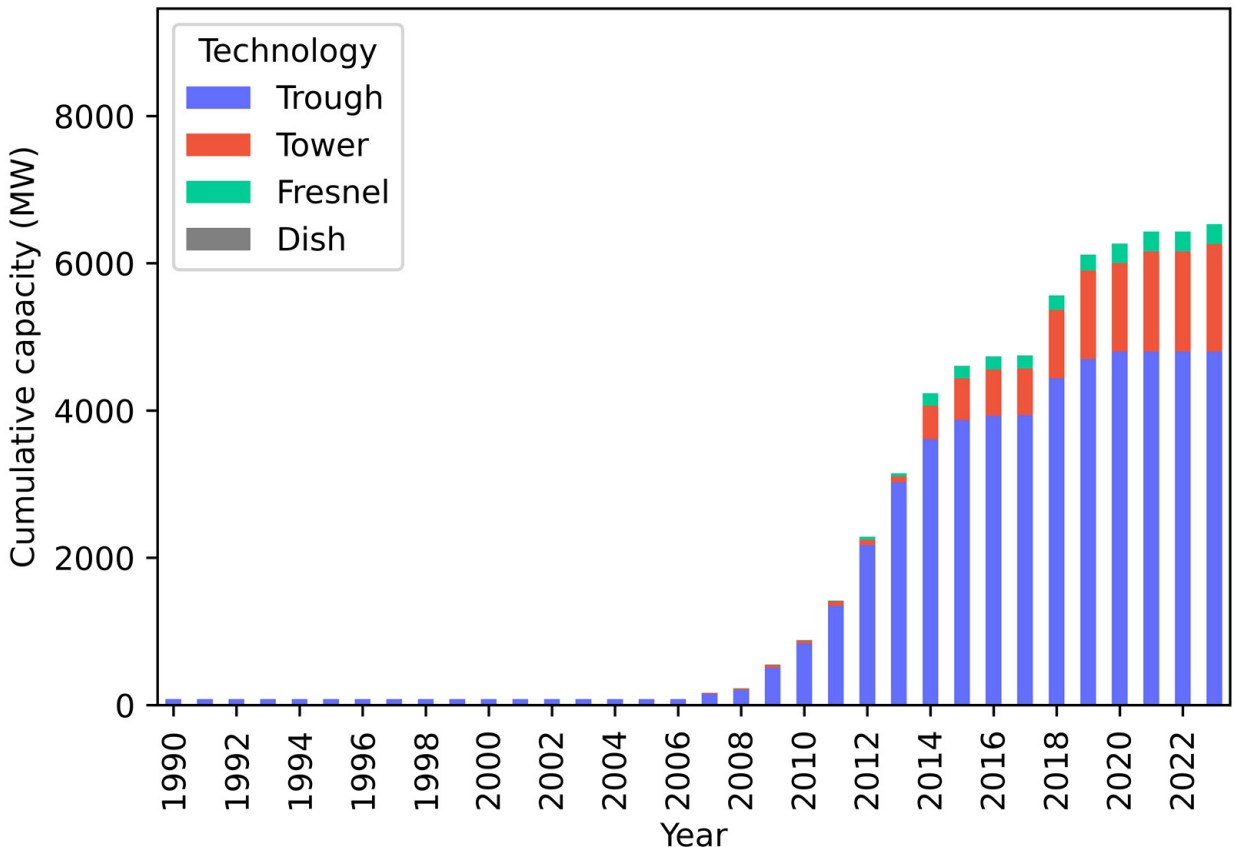

**Figure 4.** Repartition of the CSP technologies since 1990.

*2.3. Storage Capacity*

Figure 5 reports the evolution of the storage capacity of the CSP plants in hours of operation at nominal capacity. The plot distinguishes trough, tower and Fresnel technologies in blue, red and green colors, respectively. The size of the dots is proportional to the plant capacity. Only plants with reported storage capacities are plotted. The increasing storage capacities have been obvious during the last 20 years. The Gemasolar power plant, started in 2011 in Spain, demonstrated round-the-clock operation with 15 h of storage [20]. It is based on a molten salt receiver coupled with direct two-tank storage. In 2021, the Atacama I project (Chile) reached a 17.5 h storage capacity for a similar technology at a 110 MW scale. This is the largest storage capacity (in terms of hours) operated to date. Since 2020, CSP plants with storage were planned with at least 8 h of storage. Indeed, with the decreasing costs of PV technology that has become more competitive than CSP, energy storage is crucial for CSP plants. A recent study [21] shows that CSP with thermal storage could compete with PV + batteries up to 2050 for storage capacity higher than 4 h. Indeed, energy storage in batteries remains more expensive than thermal storage. Without storage, the capacity factors of CSP plants vary between 15 and 30%. When storage is added, capacity factors can reach values as high as 68% (Lanzhou Dacheng Dunhuang (DCTC Dunhuang)—10 MW Fresnel CSP Project with 16 h of storage).

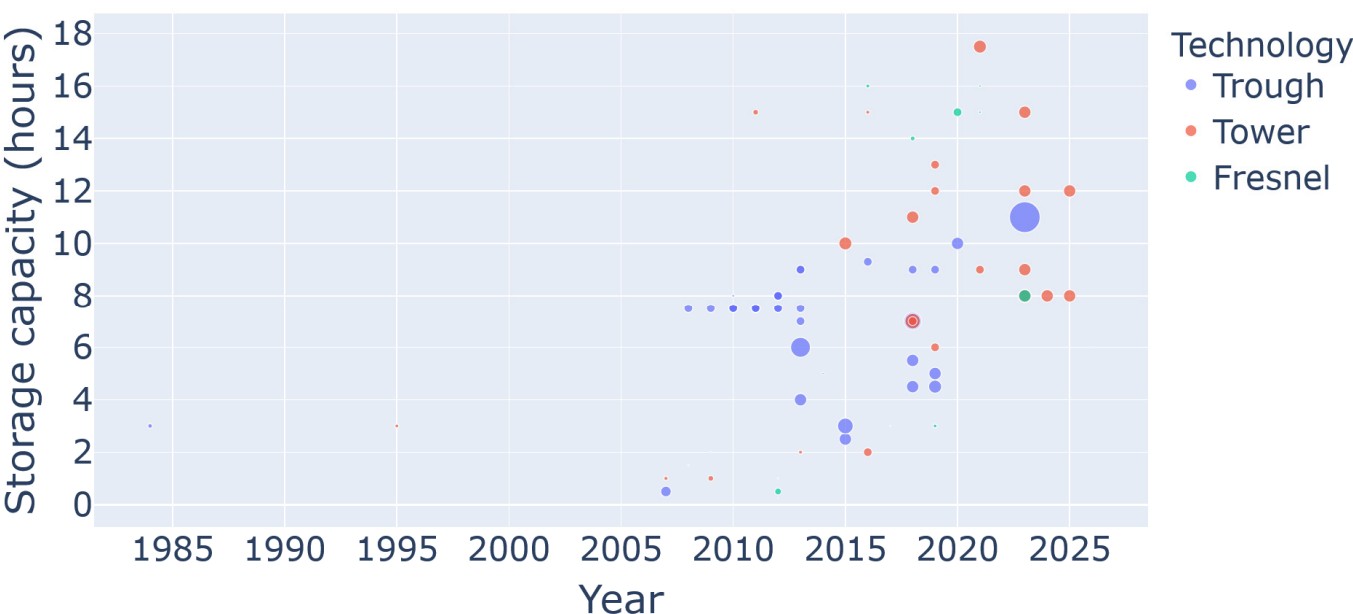

**Figure 5.** Evolution of the storage capacity of the CSP plants (in hours of operation at nominal capacity).

*2.4. Operating Temperatures*

Figure 6 plots the receiver outlet temperature of CSP plants. The plot distinguishes trough, tower and Fresnel technologies in blue, red and green colors, respectively. The size of the dots is proportional to the plant capacity. Three main temperature levels appear and depend on the heat transfer fluid used [22]. The first level at about 400 °C is directly linked to the maximum operating temperature of thermal oil in parabolic trough plants. As a parabolic trough is the more deployed technology, this is the most represented temperature level. The second temperature below 300 °C mainly corresponds to the Fresnel power plant in direct steam generation. Finally, towers reach the highest working temperatures at about 570 °C when they use molten salts. This technology has emerged during the last five years and could become a reference in the future. As an exception, two green dots (Fresnel power plant) are visible among the red ones (towers). It corresponds to the Lanzhou Dacheng Dunhuang 50 MW Fresnel power plant in China and the CSP1 Sicily Partanna MS-LFR CSP Project. These are the first Fresnel solar power plants operating at such temperatures with molten salt as heat transfer fluid. Both have a direct two-tank thermal energy storage of 15 h. In a word, the trend in the last few years was clearly to go to higher temperatures mainly with central receivers [23]. This paves the way for the use of concentrated solar energy for high-temperature chemical processes such as pyrolysis or reforming [24]. They could enable long-term energy storage in the form of chemical energy (hydrogen, biofuels, etc.).

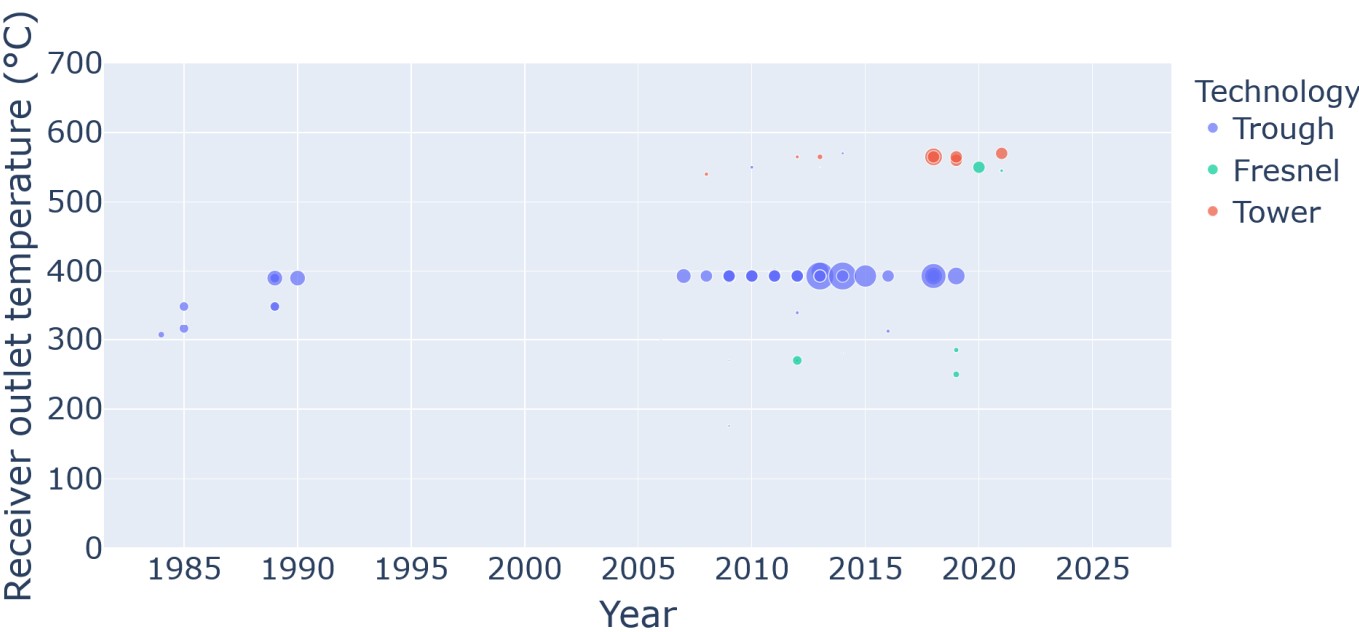

**Figure 6.** Evolution of the receiver outlet temperature of current operating CSP plants.

### 2.5. Investment Costs

Figure 7 displays the specific investment costs for the various CSP technologies (blue: trough, red: tower, green: Fresnel, brown: hybrid plants). Circles refer to CSP plants with thermal storage with the size depending on the storage capacity. Crosses refer to CSP plants without storage (concerns only the trough technology). For the economic part, hybrid plants were not dispatched to their respective CSP technology as figures are given for the whole hybrid project (CSP + PV). Consequently, hybrid projects are separated from non-hybrid projects to ensure a rigorous analysis. As already discussed, for about 10 years, most of the projects have included storage capacities. However, specific investment costs are decreasing. The lowest value (2000 USD/kWh) is from the hybrid 900 MW PV + 100 MW solar tower from Power China Ruoqiang (molten salt receiver) with 8 h of storage, which is almost one-tenth of the investment cost of the Gemasolar power plant (20 MW, 15 h of storage capacity) started in 2021. This highlights the technology's quick development and promising perspectives. However, it has to be stressed here that this lowest investment cost reported is for the hybrid plant including PV + CSP, which does not directly provide the investment cost for the CSP part only, which is a limitation of the CSP-GURU database at the moment. For a non-hybrid solar power plant, the lowest investment cost reported is for the SUPCON Delingha 10 MW Tower with an investment cost of 2635 USD/kWh.

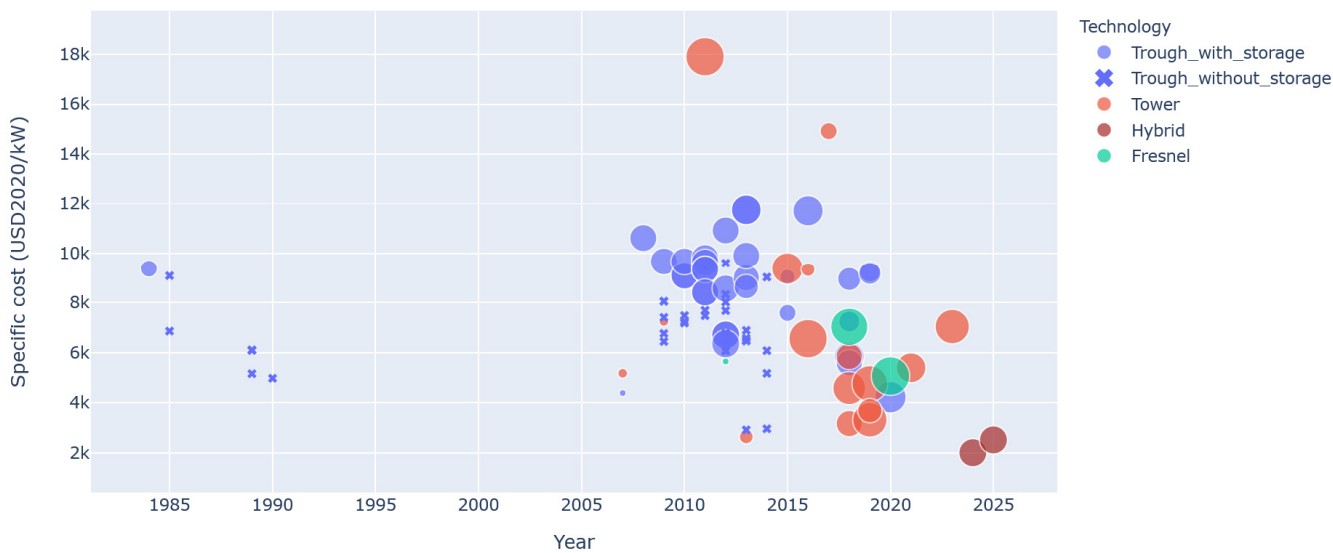

**Figure 7.** Specific investment costs for CSP plants since 1984 (blue: trough, red: tower, green: Fresnel, brown: hybrid plant). Circles refer to CSP plants with thermal storage with the size depending on the storage capacity. Crosses refer to CSP plants without storage.

*2.6. Levelized Cost of Electricity*

Figure 8 plots the available LCOE (Levelized cost of electricity with a 5% weighted average cost of capital and a 25-year payback period, capacity-dependent O&M (1.5% of investment cost per year), deflated from Year_operational using the World Bank's GDP deflator; if the station is under development or construction, then it is not deflated (assumed cost year is 2020.)) for CSP plants since 1984. Colors refer to the CSP technology while the size of each point is related to the nominal capacity of the solar power plant. As can be observed, LCOE started with values up to 0.7 USD/kWh for the decommissioned Solar Electric Generation Station II in the USA in 1985. It was operated with a 30 MW thermal oil receiver at a maximum operating temperature of 316 °C. However, current values are below 0.1 USD/kWh. The lowest value reported is for the 100 MW Shouhang Dunhuang phase II solar tower operating with molten salt in China with a value of 0.08 USD/kWh. This plant has an 11 h thermal storage capacity and started operation in 2018. The LCOE trend is obviously declining even if CSP plants include storage capacities, which increases the total investment cost. It has to be noted that the CSP-GURU database does not report the LCOE of projects after 2020, as it is generally not available, especially from recent Chinese solar plants. Interestingly, Figure 9 plots the LCOE in the same way as in Figure 8, not as a function of years but as a function of yearly DNI resources. Surprisingly, it can be seen that the lowest LCOE is found at the lowest yearly DNI (even below 2000 kWh/m$^2$/year). This is explained by the fact that these lowest LCOE projects correspond to the recent CSP supply chain developments in China [25] and India with important cost reductions. This also highlights a huge potential for cost reduction when these technologies are deployed in more sunny regions with yearly DNI superior to 3000 kWh/m$^2$/year, such as in California or the Atacama desert in Chile (LCOE of hybrid Atacama I project not available in the database).

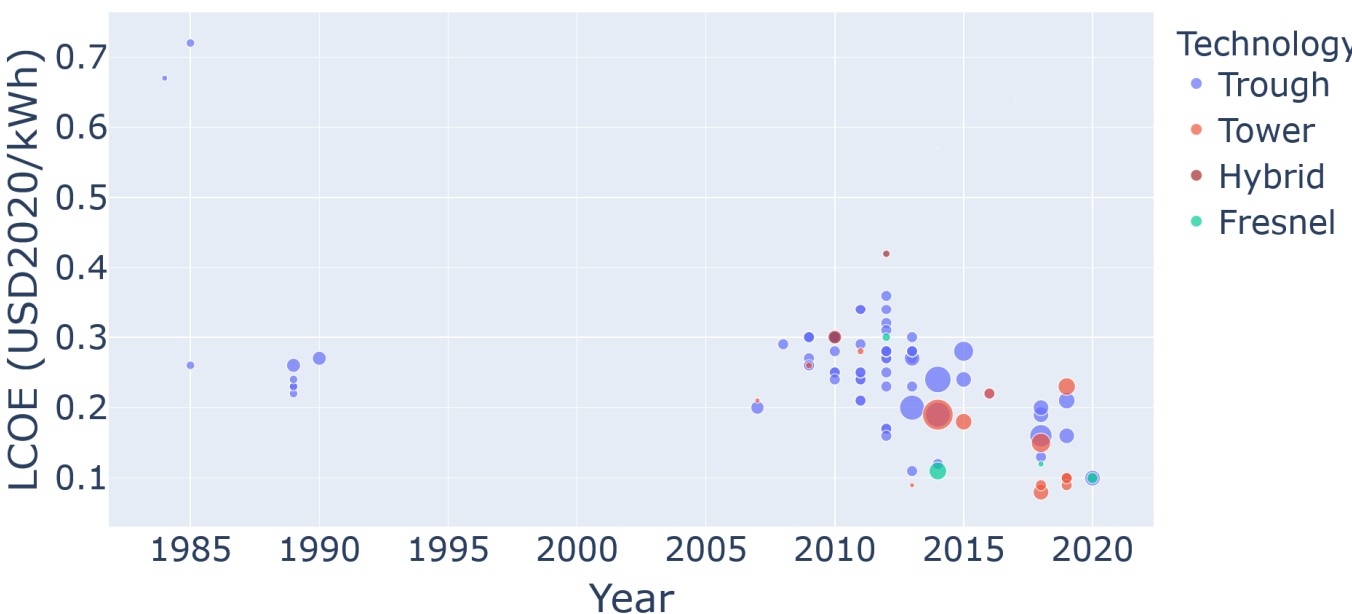

**Figure 8.** LCOE of the CSP plants since 1984.

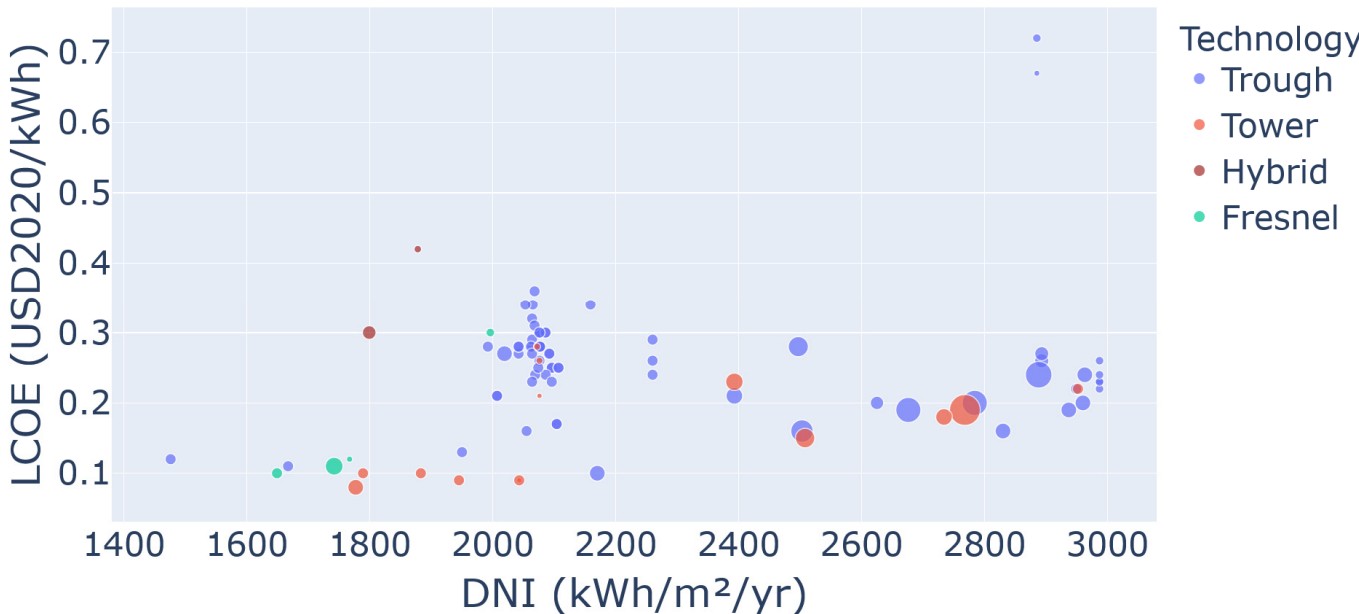

**Figure 9.** LCOE of the CSP plants depending on the yearly DNI.

### 2.7. Land Requirements

Figure 10 provides the surface ratio of the solar power plants depending on the technologies, which is also called the land use factor. It is defined as the ratio between the aperture area of the mirrors and the land used by the whole CSP station. The denser technology appears to be Fresnel (median surface ratio higher than 0.4), followed by a trough (median surface ratio about 0.25) and tower (median surface ratio below 0.2). This simply means that a solar power plant occupies from 2.5 to 5 times its mirror surface. However, this ratio is not sufficient to highlight the potential of each solar concentrating technology and Figure 11 displays the surface required per MW. This figure only considers CSP plants without thermal storage so that the mirror surface can be directly correlated to the MW produced. According to this performance indicator, the most efficient technology appears to be the Fresnel power plant with a median ratio of 5680 m$^2$ of mirror per MW,

followed by trough technology (6410 m$^2$ of mirror per MW) and towers (6692 m$^2$ of mirror per MW). This means that the Fresnel technology has a better use of the land surface and may be of importance when considering limited land availability. This can be the case for projects in constrained areas.

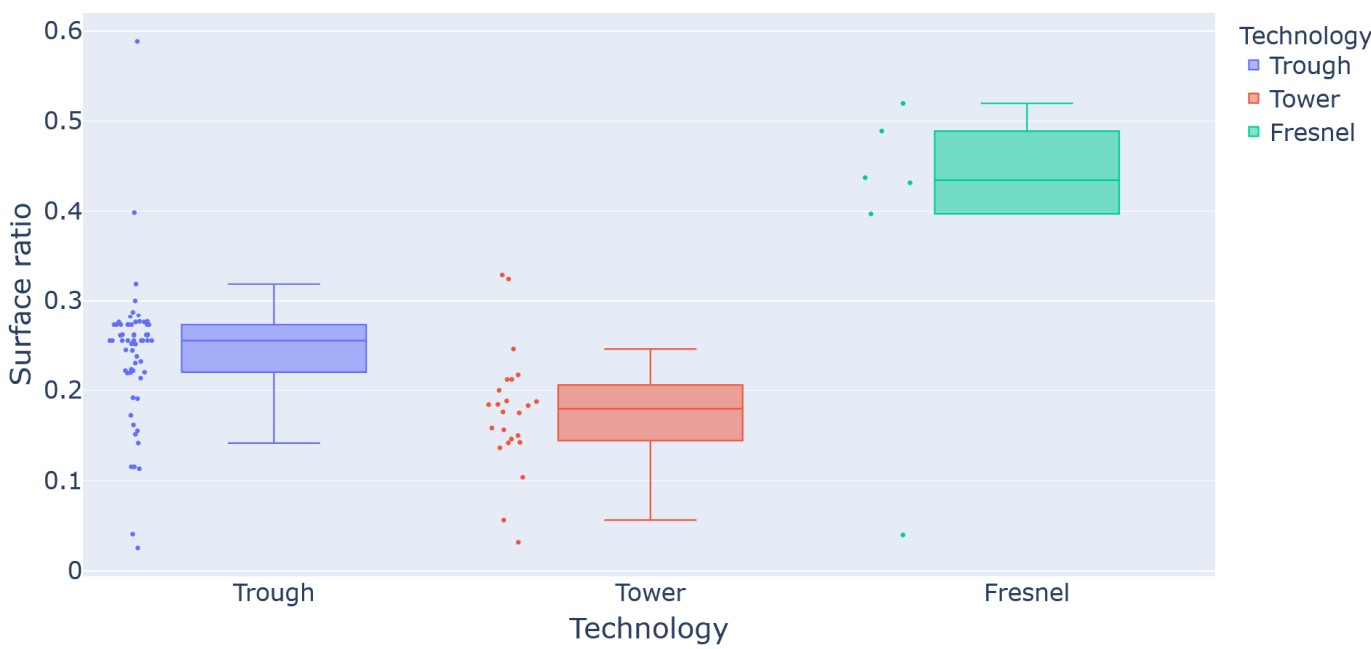

**Figure 10.** Surface ratio of the solar power plants.

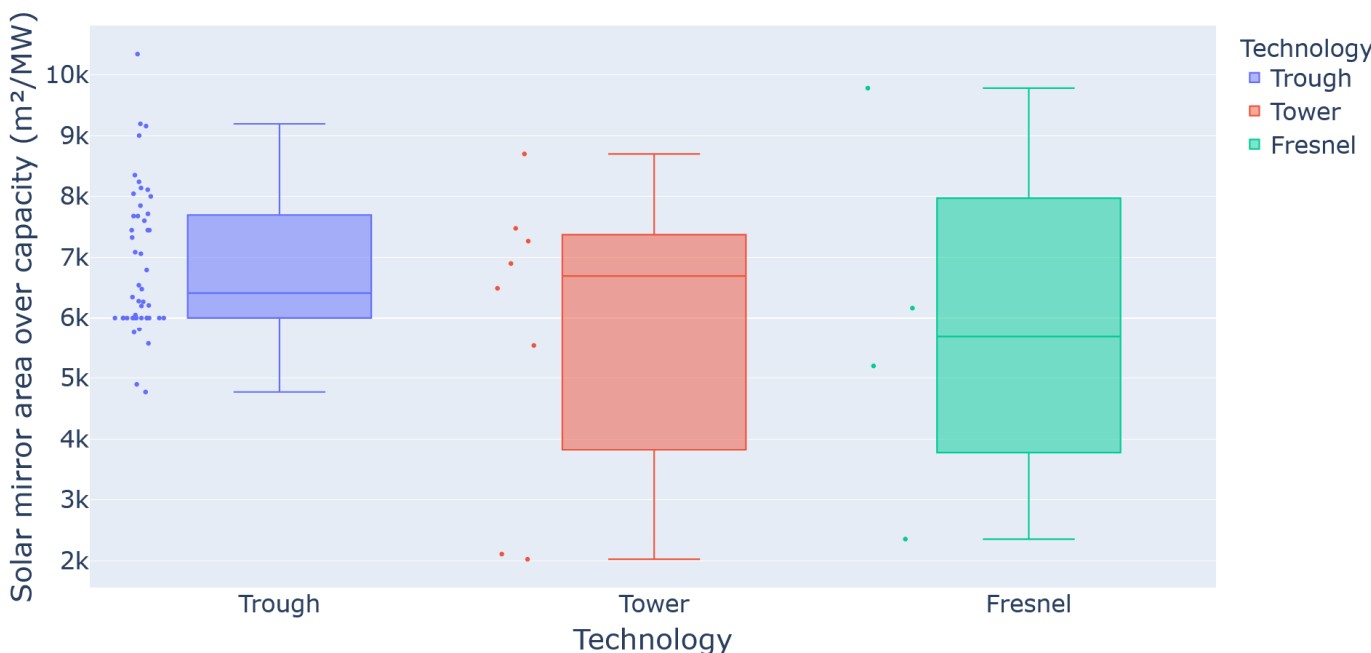

**Figure 11.** Solar mirror area required per MW (solar plants without storage).

*2.8. Overall Efficiency*

The overall efficiency may be a better indicator when land availability is no big concern (Figure 12). It is defined as the ratio between the total energy produced over the year and the product of the yearly DNI resource with the mirror area. Put another way, for one year, this is the ratio between the energy produced and the energy that would reach the

heliostat surface if it was directed normally to the sun's rays. Thus, it is more related to the efficient use of the incident solar energy. In this case, solar towers show the best overall efficiency with a median value of 15%, which is almost the same as the one for parabolic troughs. The median value for the Fresnel power plant is lower (10%). By increasing the receiver temperature, one can expect tower technology to become even more efficient as the Carnot/Rankine efficiency increases with temperature. This is one of the main levers to improve the efficiency of concentrated solar power plants [26].

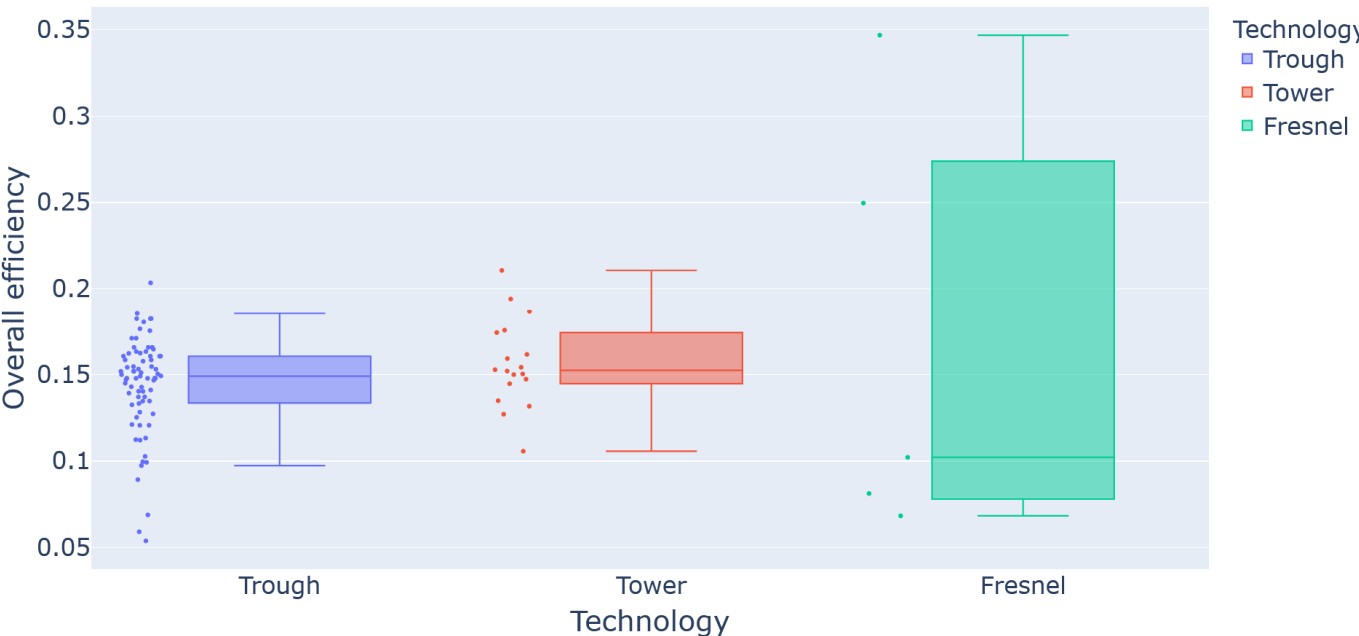

**Figure 12.** Overall efficiency of CSP plants.

### 3. Conclusions, Discussion and Perspectives

This paper draws a short review of past and present data about electricity production from concentrated solar energy. The worldwide capacity was beyond 7 GW in operation at the end of the year 2023, historically mainly in Spain and the USA but with new projects in China and UAE. Among the four available solar concentrating technologies (Fresnel, trough, tower and dish), the parabolic trough is the most represented with about 75% of the installed capacities, the rest being mainly solar towers with an increasing part over the last decade. Most of the new projects coming out include thermal storage capacities as high as 17.5 h in the form of sensible molten salt direct two-tank systems. The deployment of the central towers enables higher temperatures for the heat transfer fluid to be reached (up to 570 °C, limited by the molten salt heat transfer fluid thermal resistance). Investment costs as low as 2635 USD/kWh (non-hybrid plant) have been reached while the minimum LCOE is 0.08 USD/kWh. The Fresnel technology appears to be the one that best uses the land surface. However, tower and trough technologies offer the best overall efficiency of about 15%. This work intends also to provide a basis for other renewable energies benchmarking. Nevertheless, all the presented figures have to be interpreted with caution and awareness. Costs always depend on local solar resources and thus this point is of major importance for comparisons with other technologies (such as PV). Storage capacity is also a strong influencing parameter and technologies must always be compared for an equivalent service. The annual technology baseline from NREL provides useful information about technology-specific cost and performance parameters for electricity-generating and storage technologies, both at present and with projections through 2050 that may help to compare presented CSP data with other renewable energies [27].

The water consumption of CSP plants has to be considered carefully as most of the time sunny regions correlate with dry regions. Water consumption of the solar power plant is not often provided and is strongly depending on the plant site (dusty or not) and technology (wet or dry cooling). The work of Bošnjaković et al. [28] showed the water consumption for a solar power plant with wet cooling of 3.8 m$^3$/MWh. In the WASCOP project [29], it was shown that 90% of a solar power plant's water consumption comes from wet cooling. Going for dry cooling will thus reduce water consumption to one-tenth, which is similar to what is expected for a PV plant (about 0.5 m$^3$/MWh). The WASCOP project also draws several innovative solutions and strategies to minimize water consumption (hybrid cooling, antisoiling coatings, ultrasonic cleaners, etc.) with minimal impact on plant performance. Water consumption is thus not a strong bottleneck provided that the water local resource is taken into account when selecting technological solutions [30].

This state of the art also opens up some questions on how the technology could proceed in the next decade. Will receiver temperature continue to increase with the use of Gen3 CSP technology [31]? Will tower plant capacities overpass parabolic trough leadership? Will China continue CSP deployment along with the United Arab Emirates and disseminate low-cost solutions? Will Spain and the US be back in the CSP course? What are the outsiders to come? In the meantime, the CSP technology should be pushed forward as it is almost the only renewable energy enabled to operate round the clock. Several papers provide roadmaps and key actions for governments, industry and utilities [32–34]. CSP requires strong initial investments, so it is important for the industry to be able to foresee the market deployment and potential subsidies. To support the CSP industry, a development roadmap has to be set with coherent funding policies. These aspects were addressed recently by the European Union [35] and China [36]. The present work clearly shows the discontinuous deployment of the CSP technology, especially for Spain and the USA, which are the countries with the highest capacities but without any new projects. Also, environmental impact (life cycle assessment) has to be considered in technological choices as energy storage in batteries has more impact than thermal storage used in CSP plants [37]. A policy to support CSP would need to address the remuneration of dispatchable power in non-PV hours. Policies in China, South Africa and Morocco are promoted in this way. Moreover, it would be beneficial to build a sufficiently large pipeline of CSP projects so that automated manufacturing and economics of scale could play an increasingly role. To meet 1.5-degree compatible scenarios, a stronger expansion is needed [38].

Beyond CSP plants for electricity (on which this paper reports), CSP is also widely deployed for process heat supply. A database is also available online [39] and concentrated solar energy will most probably take an increasing share in this dynamic market given the fact that the industry requires much more heat than electricity (25% electricity need vs. 75% heat need [40]). In the future, solar energy could also be converted to solar fuels for long-term storage and long-distance transportation so that the question of solar resource temporal variability and spatial distribution would be fully tackled.

**Supplementary Materials:** The supporting information can be downloaded at: https://www.mdpi.com/article/10.3390/cleantechnol6010018/s1. HTML figures are attached to this paper to benefit from additional interactive capabilities (zoom, hover data, etc.).

**Author Contributions:** Conceptualization, S.R.; methodology, S.R. software, S.R.; validation, S.R., and R.T.; data curation, R.T.; writing—original draft preparation, S.R.; writing—review and editing, S.R. and R.T.; visualization, S.R.; supervision, S.R. All authors have read and agreed to the published version of the manuscript.

**Funding:** Data collection for CSP.guru has been supported by the IEA SolarPACES secretariat.

**Data Availability Statement:** Raw data are available at https://csp.guru/ accessed on 22 February 2024. Figure modification is possible upon request for educational and scientific purposes, provided that authors are properly referenced.

**Conflicts of Interest:** The authors declare no conflicts of interest.

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
