# Peer review of "Status of Concentrated Solar Power Plants Installed Worldwide: Past and Present Data"

_cleantechnol, doi:10.3390/cleantechnol6010018_

Round 1

Reviewer 1 Report

Comments and Suggestions for Authors

This paper presents data that are not new in sole.

But the data and statistics are represented with good demonstration and analysis view.

It will not act as a scientific or research reference as they mention in their manuscript, but it clarifies the comparison between the different CSP technologies nicely.

Reviewer 2 Report

Comments and Suggestions for Authors

This review article provides a comprehensive review of the current status of Concentrated Solar Power (CSP) technology globally, as of 2023. Leveraging data from the CSP-GURU database, the authors offer a detailed analysis of the evolution, technological trends, storage capacities, investment costs, and overall efficiency of CSP plants. The manuscript is well-organized, and the figures effectively complement the text by providing visual representations of complex data. However, authors are invited to consider some comments/recommendations as below:

1.       Abstract, page 1, line 11, “The use of this energy is expending very rapidly mainly through the photovoltaic technology” do you mean expanding instead?

2.       Page 1, line 17, avoid using words such as “we”.

3.       Page 1, line 38, “Various approaches to concentrating sun light exits (Figure 1) and can be

4.       categorized” do you mean exists instead?

5.       While the manuscript excels in detailing CSP technology's current status, it could benefit from a more explicit comparison with other renewable energy technologies, especially in terms of efficiency, cost, and scalability. This comparison would help contextualize CSP's role in the broader renewable energy landscape.

6.       The manuscript discusses investment costs and LCOE; however, a deeper economic analysis, including payback periods, financial incentives, and market dynamics, could enhance understanding of CSP's economic viability.

7.       A discussion on the environmental impact of CSP plants, including land use and water consumption, compared to other renewable energy sources, would be beneficial. This analysis could help address sustainability concerns associated with CSP technology

8.       For a review article, particularly one that aims to provide a comprehensive overview of the status of concentrated solar power (CSP) plants worldwide, including technological, economic, and policy aspects, 25 references might indeed be considered limited in scope. Review articles typically synthesize existing literature to present a state-of-the-art understanding of the subject matter, often citing a wide range of sources to ensure depth and breadth in coverage. The authors are invited to expand their reference lists, possibly by Incorporating more recent studies, reviews, and meta-analyses that have examined various aspects of CSP technology, including advancements in materials, efficiency improvements, cost reduction strategies, and integration with the broader energy grid.

9.       Based on the analysis performed in this review, what policy measures would the authors recommend to support the growth and development of the CSP industry? How can governments and international bodies best facilitate the expansion of CSP to meet global renewable energy targets? Kindly provide insights in this regard.

Comments on the Quality of English Language

Minor English editing is required

Reviewer 3 Report

Comments and Suggestions for Authors

The article reviews CSP technologies, their market, their costs and efficiencies. Overall, the text contains an interesting discussion and recent data. The authors should address the following points to further improve the quality of the manuscript: 

1. Please give more information about how the data in the excel file csp-guru.xls was collected (field visits, e-mail correspondence, from specialized magazines or internet portals, contact with local/national energy authorities). The data was obtained from what is considered the most reliable source, or you double- and triple-check it with all the other available sources of information?

2. With regard to comment #1, and from my personal experience, comparing energy data from minor markets and from different countries can be hard sometimes since different approaches are used for data collection and representation, or the data is simply not available due to confidentiality issues. For example, the SIC and LCOE data contains parameters that could change over time, and can be differently interpreted in different countries. The power generating capacities and the steam cycle efficiencies could be represented in gross or net values while the involved technologies and power plant capacities could change over time due to modifications, partial shutdowns or upgrades. How do you keep track with all these possible changes? 

3. I found that the IRENA report on Renewable Power Generation Costs in 2022 uses the CSP-guru database as a reference for writing their report. The IRENA report uses the cps-guru data directly or applies different metrics and methodologies to recalculate the CSP data from cps-guru? Did you compare the cps-guru data with that of IRENA - what are your thoughts? https://www.irena.org/Publications/2023/Aug/Renewable-Power-Generation-Costs-in-2022.

4. NREL hosts a CSP database (https://solarpaces.nrel.gov/). What are the differences between cps-guru and the NREL database?

5. Line 100: in your opinion, why Spain and USA have no CSP projects under development and the CSP market is mainly growing in UAE, China and India? State subsidies?

6. The Dubai Solar Park comprises 600 MW of parabolic trough, 100 MW of solar tower and around 1000 MW in PV by 2023? By 2030, this project is expected to increase to a total capacity of 5 GW by installing only new PV? This power plant achieved the lowest LCOE for a CSP plant with 73 US$/MWh (under a PPA deal), while the PV part achieved a LCOE of 24 US$/MWh. The CSP world record low LCOE would be achieved even if the power plant had no PVs or the large share of PVs in the solar park contributed to the record low LCOE from CSP?

https://www.c40.org/case-studies/dubai-s-mohammed-bin-rashid-al-maktoum-5-000mw-solar-park-aims-to-save-6-5-million-tco2e-annually/

7. Section 2.3.: what is the average capacity factor for the different CSP power plant technologies and how much did it increase over the last 10-15 years? How much does the thermal storage increase the capacity factor of a CSP plant?

8. Figures 5-7 should provide the reference circle sizes.

9. Lines 220-221: what is new about the recent CSP supply chain in China and India that is able to reduce the LCOE below 100 US$/MWh? 

10. Line 266: the Carnot cycle is an idealized cycle while the process in the solar tower resembles more the Rankine cycle.  The Rankine cycle efficiency also increases with the working fluid temperature.

Comments on the Quality of English Language

Minor improvements are necessary in the English language and writing style.

Round 2

Reviewer 3 Report

Comments and Suggestions for Authors

The authors supplied extensive answers to my questions and comments. The article has been improved accordingly.

Comments on the Quality of English Language

Minor editing necessary.